# Antioxidant Effect of the Ethyl Acetate Extract of *Potentilla indica* on Kidney Mitochondria of Streptozotocin-Induced Diabetic Rats

**DOI:** 10.3390/plants12183196

**Published:** 2023-09-07

**Authors:** Cinthia I. Landa-Moreno, Cristian M. Trejo-Hurtado, Jenaro Lemus-de la Cruz, Donovan J. Peña-Montes, Marina Murillo-Villicaña, Maribel Huerta-Cervantes, Rocío Montoya-Pérez, Rafael Salgado-Garciglia, Salvador Manzo-Avalos, Christian Cortés-Rojo, Juan Luis Monribot-Villanueva, José Antonio Guerrero-Analco, Alfredo Saavedra-Molina

**Affiliations:** 1Instituto de Investigaciones Químico-Biológicas, Universidad Michoacana de San Nicolás de Hidalgo, Francisco J. Múgica S/N, Morelia 58030, Michoacán, Mexico; 1419561g@umich.mx (C.I.L.-M.); 1920749g@umich.mx (C.M.T.-H.); 1315680c@umich.mx (J.L.-d.l.C.); 0618853j@umich.mx (D.J.P.-M.); 0850421k@umich.mx (M.M.-V.); 1505572g@umich.mx (M.H.-C.); rmontoya@umich.mx (R.M.-P.); rsalgado@umich.mx (R.S.-G.); smanzo@umich.mx (S.M.-A.); christian.cortes@umich.mx (C.C.-R.); 2Red de Estudios Moleculares Avanzados, Clúster BioMimic, Instituto de Ecología, A.C., Xalapa 91073, Veracruz, Mexico; juan.monribot@inecol.mx (J.L.M.-V.); joseantonio.guerrero@inecol.mx (J.A.G.-A.)

**Keywords:** antioxidant, diabetes mellitus, kidney mitochondria, oxidative stress, *Potentilla indica*

## Abstract

Diabetes mellitus (DM) is a metabolic disorder characterized by persistent hyperglycemia. This state may lead to an increase in oxidative stress, which contributes to the development of diabetes complications, including diabetic kidney disease. *Potentilla indica* is a traditional medicinal herb in Asia, employed in the treatment of several diseases, including DM. In this study, we investigated the antioxidant effect of the ethyl acetate extract of *Potentilla indica* both in vitro and on kidneys of streptozotocin-induced diabetic male rats. Firstly, phytochemicals were identified via UPLC-MS/MS, and their in vitro antioxidant capabilities were evaluated. Subsequently, male Wistar rats were assigned into four groups: normoglycemic control, diabetic control, normoglycemic treated with the extract, and diabetic treated with the extract. At the end of the treatment, fasting blood glucose (FBG) levels, creatinine, blood urea nitrogen (BUN), and uric acid were estimated. Furthermore, the kidneys were removed and utilized for the determination of mitochondrial reactive oxygen species (ROS) production, mitochondrial respiratory chain complex activities, mitochondrial lipid peroxidation, glutathione peroxidase (GSH-Px), superoxide dismutase (SOD), and catalase (CAT) activities. The in vitro findings showed that the major phytochemicals present in the extract were phenolic compounds, which exhibited a potent antioxidant activity. Moreover, the administration of the *P. indica* extract reduced creatinine and BUN levels, ROS production, and lipid peroxidation and improved mitochondrial respiratory chain complex activity and GSH-Px, SODk, and CAT activities when compared to the diabetic control group. In conclusion, our data suggest that the ethyl acetate extract of *Potentilla indica* possesses renoprotective effects by reducing oxidative stress on the kidneys of streptozotocin-induced diabetic male rats.

## 1. Introduction

DM is a chronic metabolic disorder characterized by persistent hyperglycemia, resulting directly from defective insulin action, inadequate insulin secretion, or both [1], leading to abnormalities in carbohydrate, lipid, and protein metabolism and an increase in oxidative stress (OS) [2]. Currently, DM represents a significant public health problem worldwide because it has presented at alarming levels in recent years. In 2021, a diagnosis prevalence of 537 million people was registered, and according to the IDF (International Diabetes Federation), this figure is estimated to increase to 643 million by 2030 and 783 million by 2045 [3]. Chronic hyperglycemia is a common feature in all types of this disease and is related to the long-term damage of different organs such as the kidneys, brain, eyes, peripheral nerves, and heart, leading to the emergence of chronic complications [4,5]. The chronic complications of DM include diabetic kidney disease (DKD), neuropathy, retinopathy, cardiovascular disease, stroke, and peripheral artery disease [6]. DKD is an exclusive chronic complication of diabetes in which renal microcirculation is affected, causing a series of functional and structural alterations, mainly at the glomerular level. The natural history of DKD includes a declining glomerular filtration rate, progressive albuminuria, and, ultimately, end-stage renal disease (ESRD). Metabolic changes associated with diabetes lead to glomerular hypertrophy, glomerulosclerosis, tubulointerstitial inflammation, and fibrosis [7]. DKD is the most common global cause of chronic kidney disease and ESRD [8]. Although various studies have been conducted to explain the molecular mechanisms behind the development of diabetes complications, their precise pathophysiology remains unknown. However, OS has been considered one of the key causes for the emergence of diabetes complications, including DKD. OS is defined as a disruption of redox signaling and control and is a significant upstream event in the development of diabetic complications, insulin resistance, and chronic hyperglycemia [9,10].

Pharmacological therapy for the treatment of DM and DKD is known to be available, the therapeutic objective of which consists of the regulation of blood pressure and glycemic control. However, even though many patients receive pharmacological treatment, kidney damage inevitably continues to progress [11,12], and this is probably due to the adverse side effects and, eventually, reduction in the efficiency of the drug [13]. In addition, these treatments probably exclude other determining factors in the progression of diabetic complications, such as OS. Therefore, a therapeutic alternative is the use of medicinal plants with antioxidant properties as an adjunct therapy to conventional treatment. There are multiple studies about the beneficial properties of a wide range of plants to treat DM. *Potentilla indica* is a perennial herb belonging to the Rosaceae family and native to Asia. *P. indica* is widely used in traditional Asian medicine for the treatment of leprosy, tissue inflammation, congenital fever, cancer, and DM [14]. A variety of phytochemicals have been identified in this plant, including phenolic compounds, sterols, triterpenes, and volatile oils, all of which have demonstrated a wide range of biological activities such as anti-inflammatory, antimicrobial, antitumoral, and antioxidant activities, in both in vitro and in vivo models [15,16,17,18]. Therefore, we hypothesize that the ethyl acetate extract of *Potentilla indica* reduces the OS on kidneys of streptozotocin-induced diabetic rats, improving the diabetic condition. The present study aimed to investigate the antioxidant activity of the ethyl acetate extract of *P. indica* both in vitro and on kidneys of streptozotocin-induced diabetic male rats.

## 2. Results

### 2.1. Determination of Phytochemical Compounds of the Potentilla indica Extract

To determine the preliminary bioactive compounds content present in the extract, the total content of phenolic acids, flavonoids, and terpenoids was determined using gallic acid, quercetin, and linalool as standard, respectively. As illustrated in Table 1, the ethyl acetate extract of *Potentilla indica* exhibited elevated concentrations of total flavonoids (4251.7 ± 28.9 µg quercetin/mL), followed by terpenoids (659.9 ± 65.5 µg linalool/mL), and the content of phenolic acids detected was 2.84 ± 0.1 µg gallic acid/mL.

To separate, identify, and quantify the phenolic compounds present in the ethyl acetate extract of *Potentilla indica*, we utilized an analytical technique via UPLC-MS/MS (Table 2) based on the compounds’ retention time and mass spectrometry, and the results were compared with reference standards. In this context, the major phytochemical constituents of the extract were salicylic acid (161.29 + 3.18 µg/g), ferulic acid (42.09 + 0.74 µg/g), vanillic acid (33.23 + 0.57 µg/g), t-cinnamic acid (27.92 + 0.67 µg/g), and 4-coumaric acid (27.84 + 0.71 µg/g). 

### 2.2. In Vitro Antioxidant Activity

Antioxidant properties of the ethyl acetate extract of *Potentilla indica* at different concentrations were evaluated using three different in vitro methods, and the results are presented in Table 3. At all concentrations, *P. indica* extract displayed lower DPPH scavenging activity than the antioxidant positive control. On the other hand, when compared with the anti-lipid peroxidation activity of trolox, which presented a protective impact of 82.1 ± 2.6%, in contrast with the extract at 10, 25, and 35 mg/mL, it was observed that the three concentrations showed favorable antioxidant activities (61.8 ± 0.3, 75.7 ± 4.2, and 72.2 ± 2.7%, respectively). However, the concentration of 25 mg/mL exhibited an antioxidant activity statistically similar to the control to inhibit lipid peroxidation. Furthermore, the antioxidant activity of the *P. indica* extract was identified by evaluating its reducing effect on ferric ions in vitro, the absorbance being directly proportional to the reducing power, i.e., the ability of the extract to donate e- to Fe^3+^ ions and reduce them to Fe^2+^ ions. As illustrated in Table 3, it was observed that the FRAP values of the extract at 25 and 35 mg/mL (0.1406 ± 0.03 and 0.1411 ± 0.01 A, respectively) were statistically similar to the positive control (0.1798 ± 0.04 A). 

### 2.3. Effect of the Extract on Body Weight and Biochemical Parameters

We evaluated the effect of the extract on fasting blood glucose and body weight. As indicated in Table 4, there was a significant decrease in the final body weight in the diabetic control group (271.5 ± 27.5 g) when compared to the normoglycemic control group (399.7 ± 26.4 g). However, the diabetic group treated with the extract significantly increased their body weight at the end of the treatment (315.3 ± 32.7 g) when compared to the diabetic control group (271.5 ± 27.5 g). These findings show that the ethyl acetate extract of *P. indica* significantly affected the body weight in diabetic rats. Additionally, a significant increase in fasting blood glucose (FBG) levels was noticed in the diabetic control group (489.6 ± 89.9 mg/dL) in comparison to the normoglycemic control group (81.9 ± 11.1 mg/dL). Furthermore, the diabetic group treated with the extract (381.6 ± 86.7 mg/dL) displayed statistically similar FBG levels compared to the diabetic control group (489.6 ± 89.9 mg/dL). These findings indicate that the ethyl acetate extract of *P. indica* did not have a hypoglycemic effect. The levels of BUN, creatinine, and uric acid were measured as biomarkers of kidney function (Table 4). The diabetic control group showed a marked reduction in renal function, as characterized by significant increases in BUN (49.2 ± 5.5 mg/dL) and serum creatinine levels (0.6 ± 0.05 mg/dL) when compared to the normoglycemic control group (23.7 ± 5.9 and 0.3 ± 0.02 mg/dL, respectively). Treatment with the extract under diabetic conditions significantly reduced BUN and serum creatinine (34.8 ± 4.9 and 0.3 ± 0.09 mg/dL, respectively) compared to the diabetic control group; nevertheless, the reduction in serum uric acid was not significant. 

### 2.4. Effect of the Extract on Mitochondrial Respiratory Chain Complex Activities

We evaluated the impact of the ethyl acetate extract of *P. indica* on the activities of the mitochondrial respiratory complexes (Figure 1). The activities of complexes I and II (Figure 1a,b) were increased for the diabetic control group with respect to the normoglycemic control group (43.7 and 30.0%, respectively), while the diabetic group treated with the extract showed an activity, for both complexes, significantly higher than the diabetic control group. In contrast, as illustrated in Figure 1c,d, the activities of complexes III and IV in the diabetic control group were significantly decreased with respect to the normoglycemic control group (41.2 and 21.6%, respectively). Conversely, the activities of complexes III and IV in the treated diabetic group were significantly increased compared to the diabetic control group (30.6 and 16.6%, respectively) (Figure 1c,d). These findings indicate that the ethyl acetate extract of *P. indica* enhanced the activity of the mitochondrial respiratory chain complexes in diabetic rats.

### 2.5. Effect of the Extract on ROS Production

To elucidate whether the protective effect exerted by the extract to improve the respiratory complex activities was linked with the reduction in OS, we determined the production of ROS in kidney mitochondria (Figure 2). It was observed that the ROS production of the diabetic control group (227.6 ± 29.8 ∆F) significantly increased by 63% compared to the normoglycemic control group (139.6 ± 28.6 ∆F). On the other hand, the diabetic group treated with the ethyl acetate extract of *Potentilla indica* (168.2 ± 14.6 ∆F) showed a significant decrease in ROS production compared to the diabetic control group.

### 2.6. Effect of the Ethyl Acetate Extract of P. indica on Mitochondrial Lipid Peroxidation 

To assess the protective effect of the extract on ROS-induced oxidative damage, we measured kidney mitochondrial LP by TBARS. The TBARS levels are presented in Figure 3. When compared to the normoglycemic control group, diabetic control rats had significantly higher levels of TBARS. In contrast, in diabetic rats, the treatment with the ethyl acetate extract of *P. indica* significantly decreased lipid peroxidation by ~47% compared to the diabetic control group.

### 2.7. Effect of the Ethyl Acetate Extract of P. indica on the Antioxidant Enzyme Activities

We evaluated the effect of the extract on mitochondrial SOD and GSH-Px activities. As illustrated in Figure 4a,b, the diabetic control group had significantly lower SOD and GSH-Px activity (110.2 ± 16.08 and 14.1 ± 2.05 U·mg of prot, respectively) than the normoglycemic control group (212.1 ± 43.3 and 34.4 ± 3.08 U·mg of prot, respectively). In contrast, the diabetic group treated with the ethyl acetate extract of *P. indica* (176.5 ± 16.1 and 21.4 ± 2.6 U·mg of prot, respectively) had significantly higher activity in both enzymes than the diabetic control group. We also measured the catalase activity on kidney homogenate, and the results are observed in Figure 4c. Catalase activity was significantly decreased in diabetic control group (21.5 ± 2.6 U·mg of prot) compared to normoglycemic control group (38.7 ± 4.5 U·mg of prot). On the other hand, the treatment with the ethyl acetate extract of *P. indica* enhanced catalase activity in diabetic rats (36.2 ± 2.2 U·mg of prot) in comparison to the diabetic control group.

## 3. Discussion

Diabetes is a metabolic disease involving defective insulin action, inadequate insulin secretion, or both, resulting in chronic hyperglycemia and disturbances in carbohydrate, fat, and protein metabolism [19]. Several studies have demonstrated that the diabetic state causes oxidative stress through several signaling pathways and the production of ROS, which contributes to the activation of different downstream signaling cascades, resulting in functional and structural changes in the kidney. DKD is a frequent cause of chronic kidney disease (CKD) and kidney failure, which are the major causes of morbidity and mortality associated with DM. In addition, data suggest that the elevated production of ROS and/or decreased antioxidant defenses causes OS, which is implicated in the pathogenesis of DKD [20,21]. There are currently therapeutic agents that control glucose levels in diabetic patients; however, the use of medicinal plants with antioxidant effects as adjuvant therapy, thereby slowing the progression of DKD, has been of interest.

*Potentilla indica* is a perennial plant utilized in traditional Asian medicine in which certain secondary metabolites have been identified such as ellagic acid, flavonoids, sterols, triterpenes, and volatile oils [15,16]. This plant has aroused great interest due to its diverse therapeutic activities. Considering all the above, in our present investigation, we evaluated the antioxidant effects of the ethyl acetate extract of *P. indica* both in vitro and on kidneys of STZ-induced diabetic male rats. Firstly, our in vitro results demonstrated a high content of total flavonoids present in the extract. In addition, the most abundant phytochemicals identified in the extract were phenolic acids such as salicylic acid, ferulic acid, vanillic acid, t-cinnamic acid, and 4-coumaric acid. The yield and extraction of the different phytochemicals depend on their content, polarity, and solubility and the nature of the solvents used. Ethyl acetate is a medium-polarity solvent that allows the extraction of polar, nonpolar, and intermediate-polarity compounds. A high yield of phenolic acids has been observed in extractions with polar and medium-polarity solvents. Likewise, several studies have revealed a higher content of phenolic compounds in extracts of medium and high polarity [22,23,24]; thus, these findings coincide with those reported in the literature, confirming the presence of polar and medium-polarity compounds in the ethyl acetate extract of *P. indica*, mainly phenolic compounds. Phenolic compounds are secondary metabolites produced by plants, which have been widely studied due to their pharmacological effects, highlighting their antioxidant properties [25,26]. The proposed mechanisms of action by which phenolic compounds exert their antioxidant effect include the following: (1) the neutralization of free radicals by hydrogen atom donation or electron transfer; (2) transition metal chelation; (3) suppressing enzymes associated with ROS generation; and (4) stimulating the activity of endogenous antioxidant enzymes. In this context, the presence of phenolic acids such as ferulic acid, t-cinnamic acid, and 4-coumaric acid in the extract is related to its previously observed antioxidant potential via the inhibition of lipid peroxidation and donation of e- to the ferric ion, reducing it to a ferrous ion. This antioxidant activity was statistically similar to the antioxidant positive control (Table 2). 

Subsequently, we evaluated the effect of the extract on diabetic rats. We employed a dose of 25 mg/kg due to its potent antioxidant activity in vitro and because, in our investigation group, we observed an antioxidant effect on the lungs in STZ-induced diabetic rats at that concentration. 

DM is characterized by chronic hyperglycemia. No significant difference was noticed in the glucose levels of the diabetic treated group when compared to the diabetic control group at the end of the treatment. This indicates that the ethyl acetate extract of *P. indica* does not possess a hypoglycemic effect at the concentration of 25 mg/kg. Furthermore, diabetes is associated with considerable weight loss due to increased lipolysis and proteolysis in response to chronic hyperglycemia and defective insulin action [27], leading to loss of tissue protein and increased muscle atrophy [28,29].

Oxidative stress is considered a common factor linking hyperglycemia with the complications of diabetes. Studies on experimental animal models of diabetes strongly implicate OS as a major determinant in the pathophysiology of DKD [30,31,32]. The kidney is an organ rich in oxidation reactions in the mitochondria since it requires a high energy demand to carry out its physiological functions, which makes it vulnerable to oxidative damage. Accumulating studies indicate that mitochondrial dysfunction plays a crucial role in the pathogenesis of DKD [33]. In this study, we demonstrate a significant mitochondrial Complex I hyperactivity in diabetic rats. These data are in accordance with those previously reported by Wu et al. [34,35] and Peña-Montes et al. [36]. The hyperactivity of Complex I may occur in response to the NADH/NAD+ redox imbalance caused by the overproduction of NADH under diabetic conditions and likely by post-translational 4-hydroxynonenal modifications in the NDUFS1 (75 kDa) and NDUFS2 (53 kDa) Complex I subunits [34,35]. Additionally, we observed a notable increase in Complex II activity in diabetic rats, which may be due to the availability of excess substrates, obtaining high amounts of FADH_2_ under chronic hyperglycemia conditions. However, these increases dissipated with the administration of the ethyl acetate extract of *P. indica* in diabetic rats for both complexes (Figure 1a,b). In contrast, Complex III and IV activities were reduced in diabetic rats. Alterations in Complex III activity in the kidneys of diabetic models have also been reported, and they are caused by post-translational modifications by mitochondrial ROS overproduction [37,38]. Our findings have demonstrated increased Complex I-II and decreased Complex III-IV activities, analogous with the results reported by Raza et al. [38], probably due to an increase in the reverse electron flow, leading to an increase in ROS production. In this context, we observed a significant rise in kidney mitochondrial ROS production in diabetic rats, which is probably associated with the previously observed dysfunction of the respiratory chain complex’s activities. Nevertheless, administration of the extract significantly prevented ROS overproduction under diabetic conditions (Figure 2). This may be due to the phenolic acids present in the extract, which probably neutralize the ROS generated in the mitochondria under hyperglycemic conditions, consequently reducing oxidative damage and preventing post-translational modifications in the mitochondrial respiratory complexes’ subunits. Next, we corroborated this protective effect by the extract, observing a decrease in the lipid peroxidation (Figure 3), as a biomarker of oxidative damage on the lipids of kidney mitochondrial membranes caused by excessive ROS production. These findings coincide with those previously reported by Hu et al. [39], who observed the antioxidant activity of *P. indica* in decreasing the ROS and malondialdehyde levels, a product of lipid peroxidation, assayed on an in vivo model.

In addition, diabetes-induced ROS production depletes the antioxidant defenses, leading cells to having increased susceptibility to oxidative damage [40,41]. Our findings reveal that the SOD, GSH-Px, and CAT activities in the kidneys of the diabetic control rats were significantly reduced in comparison to the normoglycemic control group (Figure 4a–c). On the other hand, the administration of the extract significantly increased the activity of antioxidant enzymes under diabetic conditions. The decline in the activities of SOD, GSH-Px, and CAT in the kidneys during a diabetic state may be due to the inactivation caused by glycosylation or by the overproduction of ROS [41,42,43]. Treatment with the ethyl acetate extract of *P. indica* increased the activity of these enzymes, probably due to the presence of phenolic acids in the extract, which have been demonstrated in numerous studies to significantly increase the expression and activity of antioxidant enzymes [44], which may be counteracting the OS caused by the ROS generated during DM.

Furthermore, BUN and serum creatinine levels in the diabetic control group displayed a significant increase compared to the normoglycemic group. However, in the diabetic rats treated with the extract, there were significant decreases in BUN and creatinine levels (Table 3). Increased BUN and serum creatinine indicates kidney damage, which leads to a decrease in the renal excretion of these biochemical components [45]. Diabetes stimulates the overproduction of ROS, causing oxidative damage, altering glomerular filtration, and increasing the permeability of the membrane [46]. This may be linked to the protective effect against OS exerted by the phenolic acids present in the extract, improving the diabetic condition and, therefore, renal function. 

According to the previous data, our research suggests that the ethyl acetate extract of *Potentilla indica* possesses phenolic acids such as ferulic acid, vanillic acid, t-cinnamic acid, and 4-coumaric acid, capable of significantly reducing oxidative stress, thus demonstrating a protective role in hyperglycemia-induced kidney damage; however, future research on this issue needs to be conducted on the majority compounds found.

In conclusion, the ethyl acetate extract of *Potentilla indica* exhibits both in vivo and in vitro antioxidant activity, mainly due to the presence of ferulic acid, vanillic acid, t-cinnamic acid, and 4-coumaric acid, which reduce serum BUN and creatinine levels, enhance mitochondrial respiratory chain complex activity, reduce lipid peroxidation and ROS production, and improve antioxidant enzyme activities (CAT, SOD, and GSH-Px) in diabetic rats (Figure 5). More investigations are required to elucidate the precise mechanisms of action involved in the renal-protective effect observed for the extracted phenolic acids from *Potentilla indica*. A limitation of our research was that experimental groups administered hypoglycemic drugs were not included. In our future studies, they can be incorporated to determine the impact of the extract under diabetic conditions when administered with a first-line hypoglycemic agent. Other limitations of the study were that studies to visualize renal morphology were not included. In future investigations, it will be necessary to include histological studies to determine whether the extract improves the anatomopathological alterations in the kidney. In addition, no other glucose homeostasis parameters were included. They may be incorporated in future research to determine whether the extract has a significant hypoglycemic effect.

## 4. Materials and Methods

### 4.1. Plant Material and Extraction 

The leaves and stems of *P. indica* were collected from the herbarium of the Instituto de Investigaciones Químico Biológicas of the Universidad Michoacana de San Nicolás de Hidalgo and were left to dry for 7 days in the absence of light. After drying, the plant material was pulverized, and the extract was obtained using successive cold maceration (4 °C) with ethyl acetate in a 1:10 ratio (p/v) for 10 days in the dark. Subsequently, it was filtered and concentrated in a rotary evaporator under vacuum at ≤60 °C; then, it was dried at ambient temperature. Finally, the extract was dissolved in mineral oil to a final concentration of 100 mg/mL, and it was stored in the dark at 4 °C until use.

### 4.2. Determination of Phytochemical Compounds of the Ethyl Acetate Extract of Potentilla indica

#### 4.2.1. Total Phenolic Acids Determination

The content of total phenolic acids was evaluated using the Folin–Ciocalteu method [47]. Briefly, 10 µL of the sample and 750 µL of the Folin–Ciocalteu solution were mixed, prepared 1:10 with sterile deionized water. The blank solution was prepared with 10 µL of ethyl acetate plus 750 µL of the Folin–Ciocalteu solution. The blends were vigorously vortexed for 5 min, then 750 µL of 6% sodium carbonate (Na_2_CO_3_) was added. Subsequently, the solutions were incubated for 60 min at ambient temperature and in the dark, to finally obtain the absorbance at 725 nm in a VELAB-VE51000UV UV-Vis spectrophotometer. The results were expressed as mg of gallic acid/mL of extract and were obtained using a standard curve of gallic acid concentrations (0–1 μmol) as a standard.

#### 4.2.2. Total Flavonoids Determination 

The method described by Kim et al., 2002, was utilized to determine total flavonoids [48]. Briefly, 10 µL of the sample was mixed with 490 µL of methanol, subjecting the mixture to vortex agitation for 1 min, later adding 1 mL of methanol, 100 µL of aluminum chloride (AlCl_3_) at 10%, and 100 µL of acetate of potassium 1 M. The samples were subjected to vortex agitation for 1 min and allowed to incubate for 30 min in the dark. The optical density was assessed at 415 nm in a spectrophotometer (UV-Vis VELAB-VE51000UV). The data are expressed in µg of quercetin/mL of extract. The quantification was obtained using a standard curve of quercetin concentrations (0–100 μmoles) as a standard. 

#### 4.2.3. Determination of Total Terpenoids

The quantification of total terpenoids was determined using the method described by Ghorai et al., 2012 [49], which consisted of mixing 1 mL of the sample with 2.5 mL of chloroform and subjecting the mix to vortex agitation for 3 min. Afterwards, it was left to stand for 10 min on ice, subsequently adding 100 µL of sulfuric acid (H_2_SO_4_). Finally, it was incubated for 90 min in covered tubes and in the absence of light. At the end of the incubation period, the reddish precipitate (presence of terpenoids) was taken from the tube using a micropipette, which was mixed with 900 µL of methanol and homogenized to subsequently obtain its absorbency. Optical density was measured at 538 nm in a spectrophotometer (UV-Vis VELAB-VE51000UV) using methanol as a blank solution. The results were reported as µg of linalool/mL of extract, using a standard curve that was made with linalool as standard (0–30 mg/mL).

#### 4.2.4. Identification and Quantification of Phenolic Compounds

The identification and quantification of phenolic compounds was performed using a 1290 infinity Agilent Ultra High-Performance Liquid Chromatography system coupled to a 6460 Agilent triple quadrupole mass spectrometer (UPLC-MS/MS), as was previously reported in Juárez-Trujillo et al. (2018) [50] and Monribot et al. (2019) [51].

### 4.3. Evaluation of Antioxidant Activity of the Ethyl Acetate Extract of Potentilla indica In Vitro

#### 4.3.1. DPPH Assay

The free-radical-scavenging activity of the extract was evaluated using a stable free radical, 2.2-dyphenyl-1picrylhydrazyl (DPPH). Briefly, the extract (10, 25 and 35 mg/mL) and the DPPH solution (0.2 mM in ethanol) were mixed. The samples were incubated for half an hour at ambient temperature in the absence of light. The absorbance was then measured at 517 nm with a Perkin Elmer Lambda 18 spectrophotometer. Trolox (500 µg/mL) was utilized as a reference standard [52].

#### 4.3.2. Anti-Lipid Peroxidation

Anti-lipid peroxidation was assayed according to Ohkawa et al. [53], with some modifications. The method is based on the extract’s ability to inhibit lipid peroxidation, a biomarker of oxidative damage, in 10% *v*/*v* egg yolk homogenate in deionized water, as a rich source of polyunsaturated fatty acids. In brief, the extract was added at different concentrations (10, 25, and 35 mg/mL) to 500 μL of egg yolk homogenate. Later, 70 mM FeSO_4_ was added to optimize the formation of malondialdehyde (MDA), and the mixture was incubated for 20 min at 4 °C. Afterwards, 2 mL of a reagent solution (15% trichloroacetic acid, 0.375% thiobarbituric acid (TBA), and 0.25 N hydrochloric acid (HCl)) was added and incubated for 20 min in a boiling water bath. After 20 min, it was warmed up and centrifuged for 5 min at 7500 rpm. After removing the supernatant, the absorbance was determined at 532 nm in a Perkin Elmer Lambda 18 spectrophotometer. A Trolox solution (500 µg/mL) was utilized as a reference standard.

#### 4.3.3. Ferric Reducing Antioxidant Power (FRAP) Determination 

The FRAP was assessed using the method given by Maruthamuthu and Kandasamy, 2016 [54]. The method relies on the ability of the extract to reduce the ferric ions (Fe^3+^) to ferrous ions (Fe^2+^). Briefly, 100 µL of the extract was diluted to different concentrations (10, 25, and 35 mg/mL) with 900 µL of deionized water. In a test tube, the diluted extract was mixed with 2.5 mL of potassium ferricyanide [K_3_Fe(CN)_6_]; 1% *w*/*w* and 2.5 mL of phosphate buffer (0.2 M, pH 6.6). Then, it was incubated in a water bath at 50 °C for 20 min. After incubation, the tube was taken from the water bath, 1.5 mL of trichloroacetic acid (10% *w*/*v*) was added, and the tube was centrifuged for 10 min at 3000 rpm. Then, 2.5 mL of the supernatant was taken, diluted with 2.5 mL of deionized water, and, finally, 0.5 mL of freshly prepared ferric chloride (FeCl_3_; 0.1% p/p) was added. The absorbance was determined at 700 nm in a Perkin Elmer Lambda 18 spectrophotometer. Trolox (500 µg/mL) was utilized as a reference standard. 

### 4.4. Animal Care and Use Statement

The animals were cared for according to the Mexican Federal Regulations for the Use and Care of Animals (NOM-062-ZOO-1999, Ministry of Agriculture, Ciudad de Mexico, Mexico), and all the study protocols were approved by the Institutional Bioethics and Biosecurity Committee of the Instituto de Investigaciones Químico Biológicas, Universidad Michoacana de San Nicolás de Hidalgo.

### 4.5. Evaluation of Antioxidant Activity of the Ethyl Acetate Extract of Potentilla indica In Vivo

#### 4.5.1. Experimental Design

Male Wistar rats (350–400 g) were used for this study. They were kept at ambient temperature with a regular 12 h light/dark cycle. Rats were provided with a standard rodent diet and water given ad libitum. The rats destined for the diabetic groups were induced 14 h after fasting with one intraperitoneal injection of 45 mg/kg streptozotocin (STZ) freshly dissolved in 0.1 M citrate buffer (pH 4.5). Only citrate buffer (vehicle) was injected into normoglycemic rats. Five days after the injection, blood glucose concentration was registered using an Accu-Check glucometer to confirm diabetes. In our study, we included only rats with blood glucose levels of more than 300 mg/dL [55]. After diabetes was confirmed, rats were randomly divided into 4 groups (*n* = 8): normoglycemic control, diabetic control, normoglycemic treated with the extract, and diabetic treated with the extract. Treatments were orally administered to the animals using an intragastric cannula once daily for 60 days with the vehicle for the control groups or with the ethyl acetate extract of *P. indica* (25 mg/kg). Glucose levels and body weight were registered initially and at the end of the treatment. All treatments were started 2 weeks after injection of STZ or citrate. Upon completion of the 60-day treatment period, the rats were fasted for 12–14 h and were euthanized. The kidneys were removed to acquire the homogenate and mitochondria, as explained below. The blood sample was obtained and centrifuged for 10 min at 5000 rpm, and the serum was separated for subsequent biochemical determinations.

#### 4.5.2. Measurement of Biochemical Serum Parameters

The glucose levels and the biomarkers of renal function, creatinine, uric acid, and BUN were determined. The serum parameters were analyzed spectrophotometrically through commercial automated equipment (DRI-CHEM Nx500i) using an enzymatic and/or colorimetric method.

#### 4.5.3. Tissue Preparation and Mitochondrial Isolation

Kidney mitochondria were isolated following the modified methodology of Saavedra-Molina and Devlin (1997) [56]. Briefly, upon completion of the treatment, rats were sacrificed by decapitation, a dissection was performed to obtain the kidneys, and the renal capsule was removed. Then, they were put into ice-cold isolation medium 1 (Mannitol 220 mM, sucrose 70 mM, Acid 3- (N-morpholino) 2 mM propanesulfonic acid (MOPS), and ethylene glycol-bis (2-aminoethyl ether)-N, N, N’N’-tetraacetic acid (EGTA) 1 mM) at 4 °C. The kidneys were cut, washed, and then homogenized using a Potter-Elvehjem homogenizer and a Teflon piston. The homogenate was centrifuged at 2000 rpm for 10 min, and then 2 mL of kidney homogenate was taken from the supernatant for future determinations; the rest was transferred to other tubes, which were centrifuged at 7500 rpm for 10 min. Subsequently, the supernatant was decanted, and the pellet (precipitate) was resuspended in isolation medium 2 (220 mM Mannitol, 70 mM sucrose, 2 mM MOPS, and 0.2% BSA (bovine serum albumin)) and centrifuged for 10 min at 10,000 rpm. Finally, the pellet was re-suspended in 1 mL of medium 2. The Biuret method was utilized to determine the concentration of mitochondrial protein using BSA as the standard [57]. 

#### 4.5.4. Determination of Mitochondrial Respiratory Chain Complex Activity

To determine the activity of mitochondrial complexes I, II, III, and IV, the kidney mitochondria were previously subjected to three continuous freeze/thaw cycles followed by osmotic shock, as mentioned by Spinazzi et al., 2012 [58]. The Complex I activity (NADH: ferricyanide oxidoreductase) was evaluated using the method reported by Peña-Montes et al., 2020 [36]. First, 0.1 mg of protein was added to 50 mM of PPB (potassium phosphate buffer) (pH 7.4) and mixed with 0.1 mg of BSA, 0.5 mM tenoyltrifluoroacetone (TTFA), 1 µg antimycin A, and 1 mM KCN. It was incubated for 7 min at ambient temperature, and then 10 µM K_3_[Fe (CN)_6_] was added as an electron acceptor. Later, the basal fluorescence was recorded for 1 min at excitation/emission wavelengths 352/464 nm in a Shimadzu RF-5301PC spectrofluorophotometer. Next, 100 µM β-nicotinamide adenine dinucleotide (β-NADH) was added, and the changes in fluorescence were recorded for 1 min. Then, 10 µM rotenone was added, and the fluorescence was assessed for 2 min. Finally, 100 µM β-NADH was added to the reaction to record the specific activity of Complex I, and changes in fluorescence were monitored for another 2 min. Specific activity was assessed using a standard curve for β-NADH. The activity of succinate-DCIP oxidoreductase (Complex II) was assessed using 0.3 mg of kidney mitochondria, resuspended in PPB (50 mM, pH 7.4), which was mixed with 0.1 mg of BSA, 10 μM rotenone, 1 mM KCN, and 1 μg of antimycin A. This blend was incubated for 7 min at ambient temperature before adding 80 M DCIP as an electron acceptor and monitoring the basal absorbance for 1 min at 600 nm. The reaction was started by adding 10 mM succinate as a substrate, and the changes in absorbance were measured for 3 min using a Perkin-Elmer UV/Vis Lambda 18 spectrophotometer. Lastly, 0.5 mM of TTFA was added to inhibit the reaction, and the absorbance was monitored for 2 min. Complex II activity was assessed using the Beer–Lambert law and the molar extinction coefficient of 21 mM^−1^ cm^−1^ for DCIP [58]. The activity of Complex II + III was assessed utilizing 0.3 mg kidney mitochondria, resuspended in PPB (50 mM, pH 7.4), followed by incubation with 0.1 mg of BSA, 10 µM rotenone, and 1 mM KCN for 7 min. Then, we added 250 µg oxidized cytochrome c as electron acceptor to the mixture, and recorded the basal absorbance for 1 min. Next, 10 mM of succinate as substrate was added, and the changes in absorbance measured for 4 min at 550 nm. The reaction was halted by adding 1 µg of antimycin A. The rate of cytochrome c reduction was assessed from the slopes of the absorbance plots using a molar extinction coefficient of 19.1 mM^−1^ cm^−1^ for cytochrome c [59]. Complex IV (cytochrome *c* oxidase) activity was evaluated using 0.1 mg of kidney mitochondria, which were incubated for 7 min with 0.1 mg of BSA, 10 µg of rotenone, 0.5 mM TTFA, and 1 µg of antimycin A. Then, 125 µg of reduced cytochrome c was added, and changes in absorbance at 550 nm were measured for 2 min using a Perkin Elmer Lambda 18 spectrophotometer. The assay was stopped by adding 1 mM of KCN [60]. Complex IV activity was evaluated utilizing the Beer–Lambert law and the molar extinction coefficient of 19.1 mM^−1^ cm^−1^ for cytochrome c [59].

#### 4.5.5. Evaluation of ROS Production 

The production of ROS was obtained by measuring the oxidation of the fluorescent probe 2′,7′-dichlorodihydrofluorescein diacetate (H_2_DCFDA). The de-acetylated form of the probe was then susceptible to oxidation, generating a fluorescent product, 2′,7′-dichlorofluorescein (DCF), which detected hydrogen peroxide, hydroxyl radical, peroxyl radical, and peroxynitrite. The accumulation of DCF indicated the production of redox-active substances [61]. Briefly, 0.3 mg/mL of mitochondrial protein was resuspended in buffer (100 mM KCl, 10 mM HEPES, 3 mM KH_2_PO_4_, and 3 mM MgCl_2_ pH 7.4) and incubated with 12.5 µM H_2_DCFDA for 15 min in an ice bath under steady shaking. Then, mitochondrial suspension was put into a quartz cuvette, and basal fluorescence was recorded for 1 min. Later, 5 mM glutamate/malate was added as a substrate and the changes in fluorescence were recorded in a Shimadzu RF-5301PC spectrofluorophotometer for 19 min at excitation/emission wavelengths of 491/518 nm.

#### 4.5.6. Lipid Peroxidation Determination

Lipid peroxidation (LP) was evaluated using thiobarbituric-acid-reactive substances (TBARSs), as reported by Buege and Aust [62], with minor changes. This method measures TBARS production, including malondialdehyde (MDA), a product of LP caused by ROS. Briefly, to avoid the interaction of the carbohydrates in the medium with the thiobarbituric acid, mitochondria were washed twice with cold PBS (pH 7.4). Following that, 0.3 mg/mL of mitochondrial protein was suspended in PBS (pH 7.4) and mixed with a reagent solution containing 0.25 M HCl, 15% trichloroacetic acid (TCA), 0.375% thiobarbituric acid (TBA), and 0.01% BHT (butylated hydroxytoluene) to avoid non-specific chromophore production. In a boiling water bath, the blend was incubated for 30 min. After cooling, the precipitate was centrifuged for 5 min at 7500 rpm. A Perkin Elmer Lambda 18 UV VIS Spectrophotometer was utilized to measure the absorbance at 532 nm, with a molar extinction coefficient of 156 mM^−1^ cm^−1^ for MDA.

#### 4.5.7. Glutathione Peroxidase Activity Determination

The glutathione peroxidase activity was evaluated following the methodology reported by Lawrence and Burk [63], with minor changes. Briefly, 0.2 mg/mL of mitochondria were resuspended in PPB 50 mM and 5 Mm EDTA and mixed with 1 mM glutathione reduced, 1 mM NaN_3_, 0.1 mg BSA, and 100 mU/mL of glutathione reductase and incubated for 5 min. At minute 4, 100 M of NADPH (β-nicotinamide adenine dinucleotide 2′-phosphate reduced) was added, and the incubation was maintained for another minute. In a Shimadzu RF5301PC, the fluorescence was measured for 1 min at excitation/emission wavelengths of 352/464 nm. Afterward, 250 M H_2_O_2_ was added, and the changes in fluorescence were recorded for 3 min at 30 °C. 

#### 4.5.8. Superoxide Dismutase Activity Determination

SOD activity was evaluated using a commercial kit (Sigma, 19160, St. Louis, MO, USA), following the manufacturer’s instructions, and calculated using the SOD from Escherichia coli as standard. The data are presented as units of SOD/mg of protein.

#### 4.5.9. Evaluation of Catalase Activity

The catalase activity was obtained in kidney homogenate, measuring the dissolved oxygen concentration, using a Clark-type oxygen electrode connected to a biological oxygen monitor (5300A Biological Oxygen Monitor, YSI, Yellows Springs, OH, USA) after adding H_2_O_2_, as mentioned by Jeulin and colleagues [64], with minor changes. Briefly, 0.2 mg of protein from homogenate tissue was resuspended in 0.1 M PPB, with 5 mM EDTA (pH 7.6), at 25 °C, and the trace was monitored for 1 min. Subsequently, freshly prepared 6 mM H_2_O_2_ was added to the chamber, and the conversion of H_2_O_2_ to oxygen was recorded for 1 min. Finally, 1.0 mM sodium azide (NaN_3_) was added to stop the reaction. Catalase activity was evaluated using bovine catalase as standard.

### 4.6. Statistical Analysis

The mean ± standard deviation was utilized to express the results. To analyze differences between groups, one-way or two-way ANOVA (analysis of variance) was utilized, followed by a Tukey’s multiple comparison test. *p*-values of ≤0.05 were regarded as statistically significant. GraphPad Prism 6 software was used to perform all data analyses.

## 5. Conclusions

The ethyl acetate extract of *Potentilla indica* exhibits both in vivo and in vitro antioxidant activity, mainly due to the presence of ferulic acid, vanillic acid, t-cinnamic acid and 4-coumaric acid, which reduce serum BUN and creatinine levels, enhance mitochondrial respiratory chain complex activity, reduce lipid peroxidation and ROS production and improve antioxidant enzymes activities (CAT, SOD and GSH-Px) in diabetic rats.

## Figures and Tables

**Figure 1 plants-12-03196-f001:**
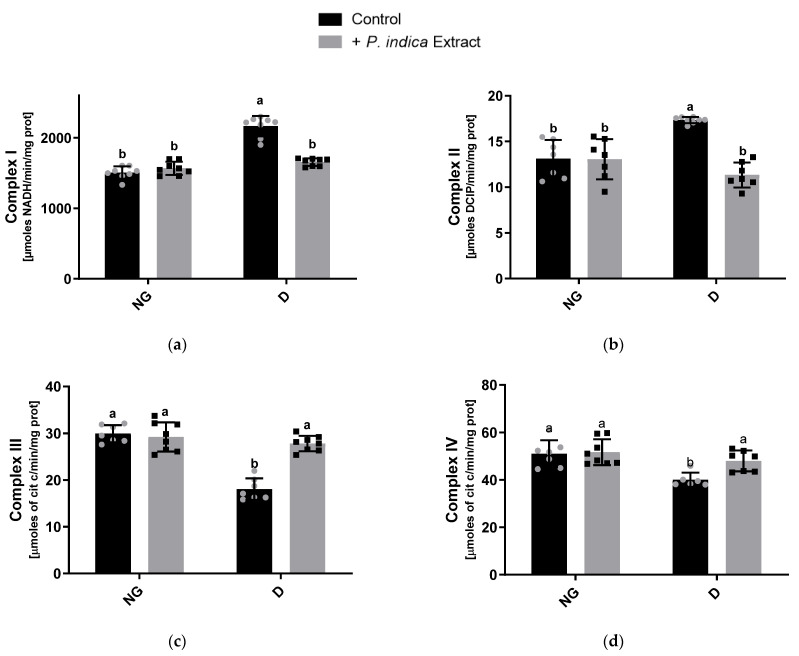
Effect of the ethyl acetate extract of *Potentilla indica* on kidney mitochondrial respiratory chain complex activities: (**a**) Complex I; (**b**) Complex II; (**c**) Complex III; and (**d**) Complex IV. Results are expressed as mean ± SD. *n* = 7–8. Two-way ANOVA, post hoc Tukey. Different letters indicate significant differences between groups, *p* < 0.05. NG: normoglycemic; D: diabetic; SD: standard deviation; grey and black symbols in the bars represent the individual data points.

**Figure 2 plants-12-03196-f002:**
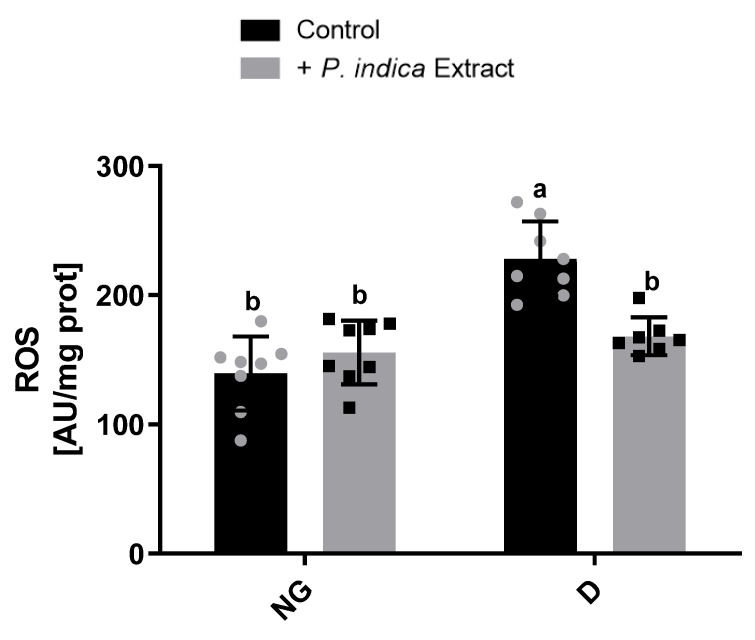
Effect of the ethyl acetate extract of *Potentilla indica* on kidney mitochondrial ROS production. Results are expressed as mean ± SD. *n* = 7–8. Two-way ANOVA, post hoc Tukey. Different letters indicate significant differences between groups, *p* < 0.05. NG: normoglycemic; D: diabetic; SD: standard deviation; grey and black symbols in the bars represent the individual data points.

**Figure 3 plants-12-03196-f003:**
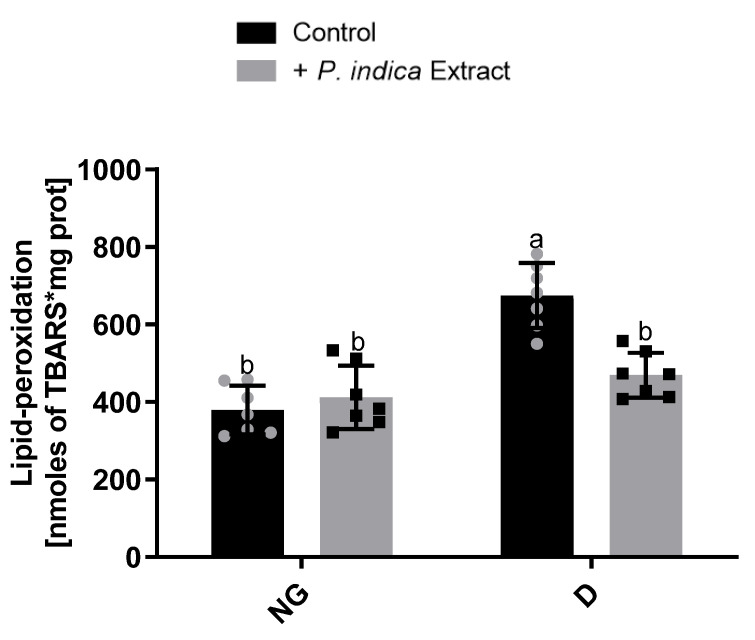
Effect of ethyl acetate extract of *Potentilla indica* on kidney mitochondrial lipid peroxidation. Results are expressed as mean ± SD. *n* = 7–8. Two-way ANOVA, post hoc Tukey. Different letters indicate significant differences between groups, *p* < 0.05. NG: normoglycemic; D: diabetic; SD: standard deviation; grey and black symbols in the bars represent the individual data points.

**Figure 4 plants-12-03196-f004:**
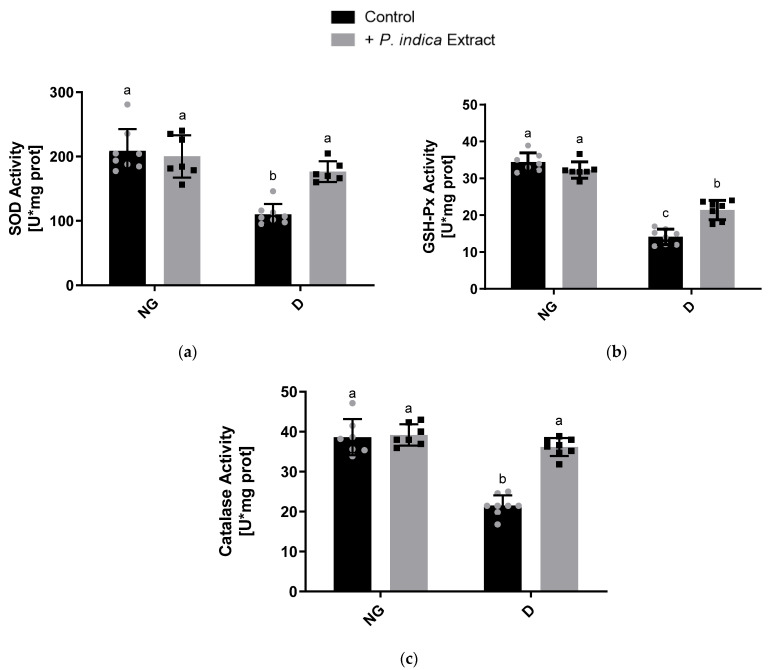
Effect of the ethyl acetate extract of *Potentilla indica* on the antioxidant enzyme activities: (**a**) superoxide dismutase; (**b**) glutathione peroxidase; and (**c**) catalase. Results are expressed as mean ± SD. *n* = 7–8. Two-way ANOVA, post hoc Tukey. Different letters indicate significant differences between groups, *p* < 0.05. NG: normoglycemic; D: diabetic; SD: standard deviation; grey and black symbols in the bars represent the individual data points.

**Figure 5 plants-12-03196-f005:**
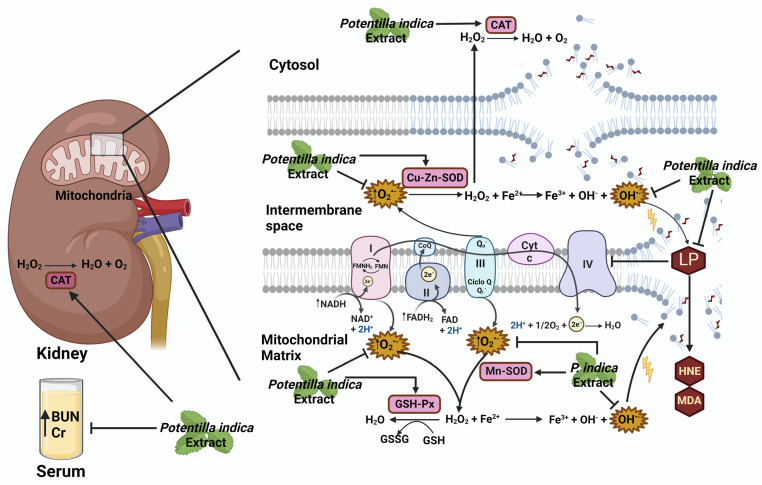
Graphical summary of results. Antioxidant effect of the ethyl acetate extract of *Potentilla indica* under diabetic conditions. Cr: creatinine; BUN: blood urea nitrogen; CAT: catalase; SOD: superoxide dismutase; GSH-Px: glutathione peroxidase; LP: lipid peroxidation; HNE: 4-hydroxynonenal; MDA: malondialdehyde.

**Table 1 plants-12-03196-t001:** Total content of phenolic acids, flavonoids, and terpenoids in the ethyl acetate extract of *Potentilla indica*.

Total Phenolic Acids (µg of Gallic Acid/mL of Extract)	Total Flavonoids (µg of Quercetin/mL of Extract)	Total Terpenoids (µg of Linalool/mL of Extract)
2.84 ± 0.1	4251.7 ± 28.9	659.9 ± 65.5

Results represent the mean ± SD of three independent experiments. SD: standard deviation.

**Table 2 plants-12-03196-t002:** Phenolic compounds present in the ethyl acetate extract of *Potentilla indica*.

Compound	Retention Time (min)	Concentration (µg/g of Dried Extract)	Compound	Retention Time (min)	Concentration (µg/g of Dried Extract)
Gallic acid	1.4	4.77 + 0.11	Scopoletin	8.4	3.43 + 0.02
Protocatechuic acid	2.5	18.46 + 0.15	Ferulic acid	8.6	42.09 + 0.74
4-Hydroxybenzoic acid	3.76	25.68 + 0.10	Salicylic acid	9.15	161.29 + 3.18
Vanillic acid	5.12	33.23 + 0.57	Ellagic acid	9.98	7.24 + 0.43
Chlorogenic acid	5.34	2.46 + 0.01	Quercetin-3-glucoside	10.26	8.54 + 0.11
Caffeic acid	5.38	13.53 + 0.23	p-Anisic acid	10.45	11.22 + 0.06
Vanillin	6.52	20.49 + 0.42	Kaempferol-3-O-glucoside	11.91	8.44 + 0.25
4-Coumaric acid	7.21	27.84 + 0.71	t-Cinnamic acid	14.08	27.92 + 0.67

Results represent the mean ± SD of three independent experiments. SD: standard deviation.

**Table 3 plants-12-03196-t003:** In vitro antioxidant activity of the ethyl acetate extract of *Potentilla indica*.

	DPPH Scavenging (%)	Anti-Lipid Peroxidation (%)	FRAP (A)
**Trolox (control)**	93.7 ± 4.1 ^a^	82.1 ± 2.6 ^a^	0.1798 ± 0.04 ^a^
**Pi Extract (10 mg/mL)**	4.2 ± 1.03 ^c^	61.8 ± 0.3 ^c^	0.0342 ± 0.01 ^b^
**Pi Extract (25 mg/mL)**	23.8 ± 0.7 ^b^	75.7 ± 4.2 ^ab^	0.1406 ± 0.03 ^a^
**Pi Extract (35 mg/mL)**	24.8 ± 2.04 ^b^	72.2 ± 2.7 ^b^	0.1411 ± 0.01 ^a^

Results represent the mean ± SD of three independent experiments according to one-way ANOVA with Tukey’s post hoc test. Different letters indicate significant differences, *p* < 0.05. Pi: *P. indica*; A: absorbance; SD: standard deviation.

**Table 4 plants-12-03196-t004:** Effect of ethyl acetate extract of *Potentilla indica* on body weight and biochemical parameters.

	Body Weight (g)		Glucose (mg/dL)		BUN (mg/dL)	Creatinine (mg/dL)	Uric Acid (mg/dL)
	Initial	Final	Initial	Final			
**NC**	342.3 ± 36.0 ^ns^	399.7 ± 26.4 ^a^	84.1 ± 7.0 ^b^	81.9 ± 11.1 ^b^	23.7 ± 5.9 ^c^	0.3 ± 0.02 ^b^	2.0 ± 0.6 ^ns^
**DC**	335.0 ± 17.2 ^ns^	271.5 ± 27.5 ^c^	468.9 ± 39.2 ^a^	489.6 ± 89.9 ^a^	49.2 ± 5.5 ^a^	0.6 ± 0.05 ^a^	3.4 ± 1.5 ^ns^
**N + Pi**	357.3 ± 13.0 ^ns^	418.6 ± 25.5 ^a^	81.8 ± 7.4 ^b^	87.5 ± 8.6 ^b^	19.1± 6.7 ^c^	0.3 ± 0.04 ^b^	1.8 ± 0.8 ^ns^
**D + Pi**	322.4 ± 41.3 ^ns^	315.3 ± 32.7 ^b^	440.8 ± 46.5 ^a^	381.6 ± 86.7 ^a^	34.8 ± 4.9 ^b^	0.3 ± 0.09 ^b^	2.2 ± 0.6 ^ns^

Data are expressed as the mean ± SD, *n* = 7–8. Different letters indicate significant differences between groups (*p* < 0.05) according to two-way ANOVA with Tukey’s post hoc test. ns: no significance; BUN: blood urea nitrogen; NC: normoglycemic control; DC: diabetic control; N + Pi: normoglycemic + 25 mg/kg of ethyl acetate extract of *P. indica*; D + Pi: diabetic + 25 mg/kg of ethyl acetate extract of *P. indica*; SD: standard deviation.

## Data Availability

The data presented in this study are available upon request from the corresponding author.

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
