# Peer review of "Antioxidant Effect of the Ethyl Acetate Extract of Potentilla indica on Kidney Mitochondria of Streptozotocin-Induced Diabetic Rats"

_plants, 2023, doi:10.3390/plants12183196_

Round 1

Reviewer 1 Report

The authors reported that crude extract of Potentilla indica possesses antioxidant activities in the kidney of diabetic rats. The results presented are preliminary and should be enhanced further with more assessments.

Introduction line 58-59: The mechanism of DM giving rise to OS and causing diabetic complications should be explained further.

Methods: Dose justification (2 mg/kg) should be done earlier.

Only rats with fasting blood glucose level > 300 mg/dL was used. Please provide a citation to this statement.

In the study design for rats, there is no positive control (standard treatment for DM), or antioxidant controls, so it is impossible to judge if the extract is superior to existing treatments or antioxidants.

The confirmation of DM is by fasting glucose alone, which I think is inadequate in this study. The use of OGTT, insulin or glycated proteins would better enhance the results.

The products of oxidation tested are limited to MDA. I would hope to see more stable products be tested, such as 8-isoprostanes and 4-Hydroxynonenal to be added.

No kidney histology? How should we assess if the kidney is injured apart from biochemical parameters?

The reporting of two-way ANOVA results should be standardized. It should start with the overall effects of DM on the parameters of interest, then the overall effects of treatment.

Results: Line 84: … exhibited elevated concentrations of total flavonoids… elevated compared to?

The limitations of this study are not mentioned. The mechanism whereby the extract exerts its antioxidant actions is not clear.

The authors should also justify the use of this extract because it does not improve the glycemic status of the rats but just improve kidney oxidative stress. 

NA

Author Response

REVIEWERS RESPONSE

 REVIEWER 1:

  • Introduction line 58-59: The mechanism of DM giving rise to OS and causing diabetic complications should be explained further.

R= The authors agree with the reviewer comment but considering that in the next paragraph (lines 60-62) relate to this sentence, we all decide to keep it.

  • Methods: Dose justification (2 mg/kg) should be done earlier.

R= The authors appreciate this comment. The dose used was 25 mg/kg (line 483), in agreement with other articles from our group (Biochemistry Research International, 2012, 603501, doi:10.1155/2012/603501; Antioxidants 2019, 8, 73; doi:10.3390/antiox8030073; J Med Food 23 (8) 2020, 827–833; Antioxidants 2023, 12, 1235. https://doi.org/10.3390/antiox12061235) and considering not using higher doses > 100 mg/kg.

  • Only rats with fasting blood glucose level > 300 mg/dL was used. Please provide a citation to this statement.

R= The authors agree with this comment. The author’s comment is that this appreciation is in relation to the articles from our group and others: Biochemistry Research International, 2012, 603501, doi:10.1155/2012/603501; Antioxidants 2019, 8, 73; doi:10.3390/antiox8030073; J Med Food 23 (8) 2020, 827–833; Antioxidants 2023, 12, 1235. https://doi.org/10.3390/antiox12061235.

  • In the study design for rats, there is no positive control (standard treatment for DM), or antioxidant controls, so it is impossible to judge if the extract is superior to existing treatments or antioxidants.

R= The authors appreciate this comment. The author’s comments on this issue are that we perform different groups (Lines 477-480) as follows: “After diabetes was confirmed, rats were randomly divided into 4 groups (n = 8): normoglycemic control, diabetic control, normoglycemic treated with the extract and diabetic treated with the extract”. So, it is included as a standard treatment for DM, which is included for all the measurements tested.

  • The confirmation of DM is by fasting glucose alone, which I think is inadequate in this study. The use of OGTT, insulin or glycated proteins would better enhance the results.

R= The authors appreciate this comment. Table 4 presents the characteristics that confirm DM status, i.e., body weight, DM with and without extract, and BUN, creatinine, and uric acid as biomarkers of kidney damage when DM, is tested with and without the extract.

  • The products of oxidation tested are limited to MDA. I would hope to see more stable products be tested, such as 8-isoprostanes and 4-Hydroxynonenal to be added.

R= The authors agree with this comment. However, Fig. 2 and Fig. 3 explain the role of mitochondrial ROS and lipid peroxidation and both results are in concordance, which suggests the direct participation of oxidative stress in DM; also, the categoric effect of diminution when added the extract of P. indica.

  • No kidney histology? How should we assess if the kidney is injured apart from biochemical parameters?

R= The authors agree with this comment of the reviewer. All the authors decide to continue these experiments but with DM type 2, project which is in progress.

  • The reporting of two-way ANOVA results should be standardized. It should start with the overall effects of DM on the parameters of interest, then the overall effects of treatment.

R= The authors agree with this comment. The reported results have been homogenized according to two-way ANOVA analysis, beginning with the effects of disease on all parameters and subsequently the effects of the treatment (line 127-146, line 156-171, line 186-190, line 203-206, line 224-228).

  • Results: Line 84: … exhibited elevated concentrations of total flavonoids… elevated compared to?

R= The authors agree with this comment of the reviewer. In Materials and Methods section in a paragraph of Total Flavonoids Determination (Line 392), it is described the quantification of total flavonoids using a standard curve of quercetin concentrations (0-100 μmoles) as a standard (Lines 399-401).

  • The limitations of this study are not mentioned. The mechanism whereby the extract exerts its antioxidant actions is not clear.

R= The authors appreciate this comment. In Figure 5, the authors add the most probable summary of the results obtained. We agree that there are more chemical reactions involved in the mitochondrial milieu, but this attempt is the first precise mechanism proposed that elucidates the action of the P. indica extract involved in renal-protective effects.

  • The authors should also justify the use of this extract because it does not improve the glycemic status of the rats but just improve kidney oxidative stress.

R= The authors agree with this comment. However, based on literature (ref. 10), “Potentilla indica is widely used in traditional Asian medicine, including leprosy, tissue inflammation, congenital fever, cancer, and DM”, as stated in the manuscript (lines 69-72), however, in our study, P. indica treatment diminished glycemia (489.6 ± 89.9) compared to diabetic control (381.6 ± 86.7).

Reviewer 2 Report

The paper by Landa-Moreno evaluated the antioxidant properties of an ehtyl extract of Potentilla indica both in vitro and using a type 1 diabetes (T1D) model, streptozotocin (STZ)-treated rats.  In addition to the abnormal characteristics consistent with T1D (blood glucose, BUN, creatine, uric acid), the authors primary focus was on parameters of oxidative stress (ROS production, enzymes involved in oxidative stress defense, mitochondrial function in the kidney).  The extract reversed or attenuated many of the aforementioned T1D-induced alterations, except for blood glucose, which was not significantly lowered by the extract in T1D rats.  Solid paper that is an more detaioled extension of earlier work on the extract.  Specific comments:

- why was the extract subject to the analysis in Table 1 and 2 when only the extract itself was used in the in vitro antioxidant analysis and T1D experiments

- Table 3 - not clear what the units of % is referring to

- it would useful for the Figures to have legends

- glucose concentrations and body weigh were measured weekly - it would be useful to show this temporal analysis and statistically evaluate the data at each time point for the 4 treatment groups.

- the Discussion would benefit from a translation of these findings to patients with T1D

Minor editing and proof reading required

Author Response

REVIEWER RESPONSE

REVIEWER 2:

  • - why was the extract subject to the analysis in Table 1 and 2 when only the extract itself was used in the in vitro antioxidant analysis and T1D experiments

R= The authors appreciate the comment of the reviewer. As far as we know, the authors determined for the first time the preliminary bioactive compounds content present in the ethyl acetate extract of P. indica. By using the proper standards of phenolic acids (gallic acid), flavonoids (quercetin) and terpenoids (linalool), that was the main reason for Table 1. Then in Table 2, it was applied the analytical technique of ULPC-MS/MS to obtain and to know the major components present.

  • - Table 3 - not clear what the units of % is referring to

R= The authors appreciate the comment of the reviewer. The percentages reported in Table 3 refer to the percent DPPH• radical scavenging activity and the percentage of inhibition of in vitro lipid peroxidation of the extract with respect to trolox as reference antioxidant.

  • - it would useful for the Figures to have legends

R= The authors appreciate this comment. The legends of figure 1 are located in line 178-180. The legends of figure 2 are located in line 196-199. The legends of figure 3 are located in line 210-213. The legends of figure 4 are in line 234-238. The legends of figure 5 are in line 363-366.

  • - glucose concentrations and body weigh were measured weekly - it would be useful to show this temporal analysis and statistically evaluate the data at each time point for the 4 treatment groups.

R= The authors agree with the reviewer comment, but since the treatment did not show a significant hypoglycemic effect over time, the authors decided to add a paragraph of the initial glycemia and at the end of the treatment (lines 283-285 and lines 482-483).

  • - the Discussion would benefit from a translation of these findings to patients with T1D.

R= The authors appreciate and agree with the reviewer comment, however, the authors consider that, hyperglycemia-induced oxidative stress may contribute to the development of chronic complications of Diabetes, including diabetic kidney disease, independently of the type of diabetes (as mentioned in line 46-49), therefore, the use of P. indica as complementary antioxidant therapy, according with our findings, could be promising for patients with diabetes mellitus in general.

Reviewer 3 Report

The study was aimed at increasing our understanding of the potential of using an antioxidant (Potentilla indica) as adjunctive therapy for diabetes mellitus. Overall, the study was interesting with logical transition from in vitro dose response testing to an in vivo model of diabetes mellitus. My questions/recommendations for improvement are as follows:

Lines 59-62. Reactive oxygen and nitrogen species are outdated terms that have been replaced with "oxidants," more appropriately categorizing these agents by their ability to accept electrons. In addition, oxidant stress is not necessarily defined as an imbalance in antioxidants and oxidant production, but more as the disruption of redox signaling and control. It is the latter when complications ensue.

2. Lines 63 - 67. You state that DM therapy is designed to control blood glucose but allude to the disease still promoting oxidant stress. Please clarify how oxidant stress remains elevated in the face of normoglycemia and leads to chronic kidney disease? Isn't the purpose of maintaining euglycemia to prevent complications to include chronic kidney disease? 

3. Lines 85 - 87. Phenolic acids are low, compared to what? The premise of your findings is based on phenolic compounds. If the concentrations are low, then could it be flavonoids or terpenoids? This is vague. In Lines 260-262 you state the most abundant phytochemicals are phenolic acids? Contradictory statements.

4. Lines 125-135. The description of body weight changes should be one to two sentences, e.g., Pi attenuated the degree of weight loss in diabetic rats. That is your finding.

5. Lines 158 - 172. You did not measure mitochondrial function (oxygen consumption rates or mitochondrial membrane potential). You measured the activity of the ETC complex. This should be stated clearly. 

6. Figures 1-4. Please show individual data points vs. means.

7. Discussion. Please avoid redundancy from introduction, results and discussion. If you state it once, there is no need to restate. This lowers the reader enthusiasm for learning something new. 

8. Lines 291-293. You did not measure muscle damage or catabolism. This should be removed. 

9. Lines 329-331. You state diabetes oxidant production depletes antioxidant defenses, yet you only show a reduction.

10. The discussion is filled with excessive speculation. Only discuss what you found. Figure 5 is a very nice depiction of how Pi may be working in diabetes. 

11. You highlight the potential of using Pi with diabetes medications, but you did not test this. A limitation that should be acknowledged. 

12. Lines 555-563. Please state which oxidants H2DCFDA detects. 

The paper should be thoroughly checked for proper English/punctation. 

Author Response

REVIEWERS RESPONSE

REVIEWER 3:

1)  Lines 59-62. Reactive oxygen and nitrogen species are outdated terms that have been replaced with "oxidants," more appropriately categorizing these agents by their ability to accept electrons. In addition, oxidant stress is not necessarily defined as an imbalance in antioxidants and oxidant production, but more as the disruption of redox signaling and control. It is the latter when complications ensue.

R= The authors appreciate the comment of the reviewer. The authors agree with the reviewer comment but considering that in the next paragraph (lines 60-62) relate to this sentence, we all decide to keep it.

2)  Lines 63 - 67. You state that DM therapy is designed to control blood glucose but allude to the disease still promoting oxidant stress. Please clarify how oxidant stress remains elevated in the face of normoglycemia and leads to chronic kidney disease? Isn't the purpose of maintaining euglycemia to prevent complications to include chronic kidney disease?

R= The authors appreciate the comment of the reviewer. However, in normoglycemia as the reviewer noticed, there is no oxidative stress in this condition. In the manuscript there is a specific sentence as follow: “therapeutic alternative is the employ of medicinal plants with antioxidant properties as an adjunct therapy to conventional treatment”, as the main purpose to use beneficial effects with the use of medicinal plants.

3)  Lines 85 - 87. Phenolic acids are low, compared to what? The premise of your findings is based on phenolic compounds. If the concentrations are low, then could it be flavonoids or terpenoids? This is vague. In Lines 260-262 you state the most abundant phytochemicals are phenolic acids? Contradictory statements.

R= The authors appreciate the comment of the reviewer. Due that this detection is semiquantitative, the phenolic acids are the quantity detected, then, part of the sentence was erased (line 86-87, and 260-262). Based on this, and according to Table 2 with the specific analytical technique of UPLC-MS/MS, the majority compound found was ferulic acid.

4)  Lines 125-135. The description of body weight changes should be one to two sentences, e.g., Pi       attenuated the degree of weight loss in diabetic rats. That is your finding.

R= The authors appreciate the comment of the reviewer.  However, a new paragraph rephrases the specific findings with the administration of the ethyl acetate extract of P. indica (lines 127-132, and lines 133-138).

5)  Lines 158 - 172. You did not measure mitochondrial function (oxygen consumption rates or    mitochondrial membrane potential). You measured the activity of the ETC complex. This should be stated clearly.

R= The authors appreciate the comment of the reviewer. A paragraph (lines 156-164, and 169-170) was added, in which it is stated that ETC activities was measured.

6)  Figures 1-4. Please show individual data points vs. means.

R= Fig. 1, complex I data points are: CNG= 1507.12±87.42mmoles/mg prot; NG + Pi= 1569.14±94.07mmoles/mg prot; CD= 2165.66±143.39mmoles/mg prot; D + Pi= 1656.32±54.23 mmoles/mg prot;  complex II data points are: CNG= 13.381±2.11mmoles/mg prot; NG + Pi= 13.065±2.2mmoles/mg prot; CD= 17.33±0.33mmoles/mg prot; D + Pi= 11.33±1.37mmoles/mg prot; complex III data points are: CNG= 30.30±1.98mmoles/mg prot; NG + Pi= 30.09±2.59mmoles/mg prot; CD= 17.80±2.83mmoles/mg prot; D + Pi= 28.54±1.01mmoles/mg prot; complex IV data points are: CNG= 51.06±5.67mmoles/mg prot; 51.67±5.46mmoles/mg prot; CD= 40.05±3.01mmoles/mg prot; D + Pi= 48.08±4.39mmoles/mg prot; Fig. 2 the data were expressed in ∆F; Fig. 3, TBARS: CNG = 379.32±66.33nmoles/mg prot; NG + Pi= 412.024±81.72nmoles/mg prot; CD= 673.8±87.69 nmoles/mg prot; D + Pi= 468.53±63.62nmoles/mg prot. Fig. 4 data points vs. means are included in the text.

7)  Discussion. Please avoid redundancy from introduction, results and discussion. If you state it            once, there is no need to restate. This lowers the reader enthusiasm for learning something new.

R= The authors appreciate this suggestion. Considering your suggestion, it was It was erased lines 126-127; 159-162; 218-220 of the manuscript.

8)  Lines 291-293. You did not measure muscle damage or catabolism. This should be removed.

R= The authors appreciate this suggestion The sentence was removed.

9)  Lines 329-331. You state diabetes oxidant production depletes antioxidant defenses, yet you only      show a reduction.

R= The authors appreciate this comment. However, the references 37 and 38 used in this manuscript, is what the authors stated. In our manuscript, the antioxidant defenses were significantly reduced (lines 328-329).

10)  The discussion is filled with excessive speculation. Only discuss what you found. Figure 5 is a very nice depiction of how Pi may be working in diabetes.

R= The authors appreciate this comment. In figure 5, the authors decide to add the most probably summary of the results obtained. We agree that there are more chemical reactions involved in the mitochondrial milieu, but this attempt is the first precise mechanism proposed that elucidate the action of the P. indica extract involved in renal-protective effects.

11)  You highlight the potential of using Pi with diabetes medications, but you did not test this. A    limitation that should be acknowledged.

R= The authors appreciate this comment. However, we did add a paragraph (lines 350-35, and line 555) explaining that this manuscript does not performed experiments with the majority compound(s) obtained from the extract of P. indica.

  • Lines 555-563. Please state which oxidants H2DCFDA detects.

R= The authors appreciate this comment. It was added to the manuscript a sentence (lines 552-555) that state how this probe reacts with ROS, which detects peroxide hydrogen.

Reviewer 4 Report

“Antioxidant Effect of the Ethyl Acetate Extract of Potentilla indica on Kidney Mitochondria of Streptozotocin-Induced Diabetic Rats” by Landa-Moreno et al. is a well-designed study and adds significant information to the field of diabetes. The methodology regarding the induction of diabetes and the treatment after the induction adheres to the respective well-established protocols. In general, the study is focused, well-designed, and well-written and deserves publication. I believe that some minor grammatical errors will be corrected through the proof reading process. I would suggest the authors to investigate the biochemical pathways of glucose metabolism in order to enlighten the underlying molecular mechanisms.

Minor editing of English language required

Author Response

There is no reviewer 4

Round 2

Reviewer 1 Report

Thank you for your reply. However, most of the previous comments were dismissed, not incorporated into the text or added to the limitations of the study. I feel that the revision is not satisfactory.

The language is satisfactory

Author Response

REVIEWER RESPONSE SECOND ROUND

REVIEWERS RESPONSE

 REVIEWER 1:

  • The limitations of this study are not mentioned. The mechanism whereby the extract exerts its antioxidant actions is not clear.

R= The authors appreciate this comment. It was included a paragraph on this issue (lines 353-359). Also, in figure 5, the authors decide to add the most probably summary of the results obtained.

Reviewer 3 Report

Lines 60-62: Once again, oxidant stress is disruption of redox signaling and control, not imbalance in antioxidant defenses and production. Figure 4 highlights this as glutathione peroxidase was still lower in the D+Pi group despite normalized oxidants. Select another reference and re-word throughout.

65-67: As stated in my first review, if you normalize glycemia, oxidant production is reduced. You state that DM medications treat blood glucose dysregulation but exclude other factors in DM complications such as oxidant stress. Again, if you have euglycemia you do not have oxidant stress assuming absence of coexisting pathologies. This fact lowers enthusiasm for this work.

Lines 125-135: Body weight description is wordy and unclear. Simplify.

Figures: Show individual data points in your figures. It is the norm to show each data point.

Lines 391 - 395: As previously stated, you do not show a depletion in your model. State this. Also, glutathione peroxidase was still reduced. You fail to mention this. You did not measure all of the antioxidants.

You do not state that a major limitation of this work is that you did not treat the DM animals for glycemia. Another study must be conducted to examine the impact of Pi on the same parameters when first line Rx's are given. 

H2DCFDA does not only indicate H2O2, but also hydroxyl radical, peroxyl radical, and peroxynitrite. Thus, it is a non-specific oxidant detector. Amplex red would have enabled you to detect just H2O2. Please correct.

The manuscript needs extensive editing. 

Author Response

REVIEWER RESPONSE SECOND ROUND

REVIEWER 3. SECOND ROUND

  1. Lines 60-62: Once again, oxidant stress is disruption of redox signaling and control, not imbalance in antioxidant defenses and production. Figure 4 highlights this as glutathione peroxidase was still lower in the D+Pi group despite normalized oxidants. Select another reference and re-word throughout.

R: The authors appreciate the comment of the reviewer. The text has been modified in the line 61 according to the suggested comments and we have incorporated the reference number 9.

  1. 65-67: As stated in my first review, if you normalize glycemia, oxidant production is reduced. You state that DM medications treat blood glucose dysregulation but exclude other factors in DM complications such as oxidant stress. Again, if you have euglycemia you do not have oxidant stress assuming absence of coexisting pathologies. This fact lowers enthusiasm for this work.

R: The authors appreciate the comment of the reviewer. Although the development of ND involves a series of metabolic and hemodynamic disorders, all of these converge in oxidative stress. In addition, there is evidence that suggests that despite the hypoglycemic and hypertensive drug treatment that patients receive, the renal damage induced by the diabetic condition inevitably continues to progress, suggesting the participation of other determining factors such as OS (Ignudi et al., 2016; Hernández et al., 2022). For this reason, this study is focused on the reduction of oxidative stress under hyperglycemia conditions to exert a protective effect on the kidney, an organ highly susceptible to oxidative damage, to prevent or delay the development of diabetic nephropathy and its progression to end stage renal disease. Therefore, addressing oxidative stress with antioxidant phytochemicals in conjunction with conventional therapy may be an important approach to improve patient prognosis and reduce diabetic mortality.

According to the above, we have modified the lines 64-69.

  1. Lines 125-135: Body weight description is wordy and unclear. Simplify.

R: The authors appreciate the comment of the reviewer. The description of body weight results has been modified in the lines 125-132 according to the suggested comments.

  1. Figures: Show individual data points in your figures. It is the norm to show each data point.

R: The authors appreciate this comment. Individual data points have been incorporated into the figures.

  1. Lines 391 - 395: As previously stated, you do not show a depletion in your model. State this. Also, glutathione peroxidase was still reduced. You fail to mention this. You did not measure all of the antioxidants.

R: The authors appreciate this comment. The activity of the antioxidant enzymes in our experimental diabetic model showed a significant decrease in comparison with the control rats. On the other hand, although the activity of glutathione peroxidase did not normalize as in the other antioxidant enzymes, its activity increased after administration of the extract in comparison with the diabetic control group. However, the mechanism by which the extract induces the activity of these enzymes under diabetic conditions is currently under investigation and the measurement of other antioxidants is also contemplated.

  1. You do not state that a major limitation of this work is that you did not treat the DM animals for glycemia. Another study must be conducted to examine the impact of Pi on the same parameters when first line Rx's are given. 

R: The authors appreciate this comment. We have added the following study limitations in lines 355-358. “A limitation of our research was that experimental groups administered with hypoglycemic drugs were not included. In future studies they can be incorporated to determine the impact of the extract under diabetic conditions when administered with a first-line hypoglycemic agents. Another limitation was the treatment period. A long-term treatment with the extract is necessary to determine if the extract has a significant hypoglycemic effect.

  1. H2DCFDA does not only indicate H2O2, but also hydroxyl radical, peroxyl radical, and peroxynitrite. Thus, it is a non-specific oxidant detector. Amplex red would have enabled you to detect just H2O2. Please correct.

R: The authors appreciate this comment. The text was corrected in the lines 555-556.

Round 3

Reviewer 1 Report

The limitations must also cover more glucose homeostasis parameters, such as OGTT, insulin, HOMA-IR and HbA1c to validate the hypoglycemic effects of the herb.

Since the kidney is the focus of this study, the lack of morphological studies of kidneys is very alarming. The authors refuse to address this in the limitations or include the results.

The language is ok

Author Response

REVIEWER RESPONSE THIRD ROUND

REVIEWERS RESPONSE

 REVIEWER 1: The limitations must also cover more glucose homeostasis parameters, such as OGTT, insulin, HOMA-IR and HbA1c to validate the hypoglycemic effects of the herb.

R= The authors agree with the reviewer’s question. A paragraph was added to address this issue on lines 362-364.

Since the kidney is the focus of this study, the lack of morphological studies of kidneys is very alarming. The authors refuse to address this in the limitations or include the results.

R= The authors agree with the reviewer’s question. A paragraph was added to address this issue on lines 360-364.

Reviewer 3 Report

Regarding secondary complications in face of normoglycemia. Please explain how you still have oxidant stress production with euglycemic agents that still lead to CKD? How. If you correct hyperglycemia, how is CKD still occurring? 

Should be improved.

Author Response

REVIEWER RESPONSE THIRD ROUND

REVIEWER 3. THIRD ROUND

REVIEWER 3: Regarding secondary complications in face of normoglycemia. Please explain how you still have oxidant stress production with euglycemic agents that still lead to CKD? How. If you correct hyperglycemia, how is CKD still occurring?

R= The authors agree with the reviewer’s request. A paragraph was added to address this issue on lines 63-69, with new references, too.
